# Evaluation of the Impact of a Midwife-Led Breastfeeding Group Intervention on Prevention of Postpartum Depression: A Multicentre Randomised Clinical Trial

**DOI:** 10.3390/nu16020227

**Published:** 2024-01-10

**Authors:** Isabel Rodríguez-Gallego, Rafael Vila-Candel, Isabel Corrales-Gutierrez, Diego Gomez-Baya, Fatima Leon-Larios

**Affiliations:** 1Foetal Medicine, Genetics and Reproduction Unit, Virgen del Rocío University Hospital, 41009 Seville, Spain; isroga@cruzroja.es; 2Red Cross Nursing University Centre, University of Seville, 41013 Seville, Spain; 3Faculty of Health Sciences, Universidad Internacional de Valencia (VIU), 46002 Valencia, Spain; 4La Ribera Primary Health Department, 46600 Alzira, Spain; 5Foundation for the Promotion of Health and Biomedical Research in the Valencian Region (FISABIO), 46020 Valencia, Spain; 6Surgery Department, Faculty of Medicine, University of Seville, 41009 Seville, Spain; 7Foetal Medicine Unit, Virgen Macarena University Hospital, 41009 Seville, Spain; 8Department of Social, Developmental and Educational Psychology, Universidad de Huelva, 21007 Huelva, Spain; diego.gomez@dpee.uhu.es; 9Nursing Department, School of Nursing, Physiotherapy and Podiatry, University of Seville, 41009 Seville, Spain; fatimaleon@us.es

**Keywords:** breastfeeding, support group, lactation, self-help group, postpartum depression, general self-efficacy, women’s mental health

## Abstract

Postpartum depression is a significant health issue affecting both mothers and newborns during the postpartum period. Group support interventions during this period have proven effective in helping women cope with depression and improving breastfeeding rates. This study aimed to assess the effectiveness of a midwife-led breastfeeding support group intervention on breastfeeding rates, postpartum depression and general self-efficacy. This was a multicentric cluster randomised controlled trial with control and intervention groups and was not blinded. It was conducted in Andalusia (southern Spain) from October 2021 to May 2023. A total of 382 women participated in the study. The results showed a significant difference in exclusive breastfeeding rates at 4 months postpartum between the groups (control 50% vs. intervention 69.9%; *p* < 0.001). Additionally, there was a lower mean score on the Edinburgh Postnatal Depression Scale in the intervention group (12.49 ± 3.6 vs. 13.39 ± 4.0; *p* = 0.044). Similarly, higher scores of general self-efficacy were observed among breastfeeding women at 2 and 4 months postpartum (77.73 ± 14.81; *p* = 0.002 and 76.46 ± 15.26; *p* < 0.001, respectively). In conclusion, midwife-led breastfeeding support groups enhanced self-efficacy, prolonged breastfeeding and reduced postpartum depression 4 months after giving birth.

## 1. Introduction

The postpartum period entails significant physical, psychosocial and social changes for women as they adapt to a new situation. Therefore, it is known as a period of special vulnerability related to maternal mental health [1]. Approximately 9.6% to 19.2% of mothers experience a major or minor depressive episode during the first 12 months after childbirth [2]. Thus, one of the main complications during the postpartum period is postpartum depression (PPD) [3,4].

Globally, one in five women is estimated to develop PPD. However, the prevalence of PPD varies significantly between geographic areas and cultures. Southern Africa has the highest reported prevalence (39.96%), eastern Europe (16.62%) and southern Europe (16.34%) show intermediate prevalence and Oceania (11.11%) has some of the lowest reported figures [5,6,7]. Furthermore, countries with higher income and developed countries have a significantly lower prevalence than lower income or developing countries [5]. However, these figures may underestimate the true extent of the problem due to barriers to detection and the stigma associated with mental illnesses in the perinatal context. Some estimates suggest that more than 50% of women with PPD are not diagnosed [6]. PPD generally occurs within 4 weeks after delivery and can last 6 months or longer after delivery, although some authors indicate that it could last up to 2 years after delivery [4,8,9,10].

Breastfeeding provides multiple demonstrated benefits on the physical, cognitive and social levels for both the mother and the newborn [11,12,13]. However, the psychological benefits, especially those concerning PPD, are still largely unknown. There is a complex physiological relationship between breastfeeding and PPD. During pregnancy, lactation begins with an increase in progesterone and estrogens that prepares the breast ducts as part of the stimulation process, but in the first days after delivery, there is a rapid decrease in both that signals the start of milk production. This rapid drop in progesterone and estrogen is a potential catalyst for the onset of mood lability and therefore PPD [14]. Progesterone derivatives (pregnenolone and allopregnanolone) target their effect in regions of the brain related to processing emotions. Establishing the exact role of these progesterone derivatives in the development of PPD treatment may enlighten a new perspective on the general pathophysiology of mood disorders because allopregnanolone interacts with GABA-A receptors and has significant anti-depressant, anti-stress, sedative and anxiolytic effects [15]. Some studies indicate that depression during pregnancy and postpartum is one of the factors that can contribute to breastfeeding failure. Other studies also suggest an association between breastfeeding and PPD, suggesting that PPD can reduce breastfeeding rates and that breastfeeding can decrease the risk of PPD. Additionally, there is evidence that breastfeeding can prevent PPD or help symptoms to recede more quickly. However, the direction of this association is still uncertain [16,17].

Due to all these reasons, PPD has become a significant health issue that affects not only women’s health by increasing maternal morbidity and mortality but also a newborn’s feeding patterns and, consequently, behavioural, emotional and cognitive development during early childhood [5,18].

Group interventions during the postpartum period, during which women share a safe space of mutual acceptance and understanding, have proven effective in improving depressive symptoms and empowering women to cope with their situation [19]. Additionally, there are also encouraging results demonstrating that group interventions are effective at maintaining breastfeeding during the postpartum period, especially when this peer support is combined with the leadership of a healthcare professional or an International Board Certified Lactation Consultant (IBCLC) [20]. Likewise, there is evidence of the positive impact that breastfeeding has on women’s mental health by enhancing their well-being, increasing perceived self-efficacy and promoting interaction with the newborn [21,22].

At the individual level, affective characteristics, or the “qualities that represent the typical ways of feeling of individuals”, are particularly important determinants of breastfeeding practices [23]. One of these key affective characteristics is self-efficacy, defined by Bandura [24] as “the belief in one’s capabilities to organise and execute the courses of action required to produce certain achievements or results”. In contrast, low levels of self-efficacy have been shown in previous studies to be a risk factor for the development of PPD [25].

Thus, the mental health of the mother constitutes a significant underlying factor linked to barriers and reduced rates of intention, initiation and maintenance of breastfeeding. Given the evidence of a bidirectional association between maternal mental health and breastfeeding, it is essential to consider both aspects when evaluating the effectiveness of interventions aimed at improving these outcomes [16,17,18,19,20,21,22,23,24,25,26,27].

The principal aim of this study was to assess the effectiveness of a midwife-led breastfeeding support group intervention on the maintenance of breastfeeding, the prevention of PPD and on general self-efficacy. Additionally, the study aimed to explore the relationship between maternal depression and breastfeeding success.

## 2. Materials and Methods

### 2.1. Study Design

This was a multicentric cluster randomised controlled trial with a control group (CG) and an intervention group (IG) and was not blinded. This study was conducted according to the latest Consolidated Standards of Reporting Trials 2010 guidelines for reporting randomised controlled trials [28] and was completed as described in our published protocol [29]. Prior to the start of the trial, it was registered in the International Standard Registered Clinical/Social Study Number registry (Trial ID: ISRCTN17263529; date recorded: 17 June 2020).

### 2.2. Participants and Study Area

Women who met the eligibility criteria were enrolled as participants from primary health centres in Andalusia, Spain. Andalusia is an autonomous community with a birth rate of 7.72 per 1000 inhabitants (2021) [30] and 4,328,407 women of reproductive age [31] with the average age at which the first child is born being 32.7 years [32]. The study involved populations from the provinces of Seville, Cadiz, Huelva, Granada and Jaen.

### 2.3. Inclusion and Exclusion Criteria

The inclusion criteria included the following:Healthy women performing exclusive or partial breastfeeding 10 days after birth and who attended antenatal lessons at the primary health center;Women over 18 years of age;Women who accepted and signed the informed consent form.

Exclusion criteria included the following:


Human immunodeficiency virus-positive;Cancer;Tuberculosis infection;No intention to breastfeed;Impossibility or contraindication to breastfeed due to medical conditions;Premature and/or complicated labour or newborn in a neonatal intensive care unit during the first month of life;Communication difficulties due to language barriers.


### 2.4. Sample Size

According to 2021 data from the National Statistical Institute of Spain, there were a total of 65,650 births in Andalusia. Specifically, the provinces of Seville (15,655 births), Granada (7083), Huelva (4227), Jaen (4499) and Cadiz (8904) accounted for 40,368 births, constituting 61.79% of the total births in the region [33]. The rate of exclusive breastfeeding (EBF) at 6 months in Andalusia is 39% [34], which was considered the baseline value in the CG. An anticipated increase of 10%, as suggested by previous research [35,36], in the rate of EBF at 6 months was established. To achieve this difference between the two groups, a two-tailed hypothesis was posed, with a power of 80% and allowing for a type I error of 5%. The necessary sample size amounted to 371 women distributed between the two study groups.

### 2.5. Randomisation and Recruitment

Primary health centres were randomly assigned to either the IG or the CG (receiving usual care), considering whether any form of group breastfeeding support intervention was already available. The allocation of health centres into these groups was performed by a research technician, who was independent of the researchers responsible for participant recruitment, using a random sequence [37]. The technician provided random unique identifiers to the health centres, distinguishing between those belonging to the CG and IG.

Subsequently, the women were again randomised following a simple strategy (1:1) at 35–37 weeks of gestation by the collaborating primary health centre midwives. Finally, each participant received an identification code based on the group to which she was assigned.

### 2.6. Intervention

Participants in the CG received standard care in terms of maternal education and postpartum visits, following the guidelines outlined in the Protocol for Care during Pregnancy, Childbirth and Puerperium by the Andalusian Health and Social Welfare Council [38], similar to the women in the IG. Within the initial 10 days after giving birth, they underwent a one-on-one visit with the midwife to address individual concerns. Additionally, women had the opportunity to request individual postpartum consultations with the designated midwife at their health centre as needed.

Women in the IG received the usual prenatal and postpartum care, just like those in the CG. Subsequently, they engaged in monthly 2 h in-person and/or virtual group sessions known as breastfeeding support groups, during which the midwife assumed the roles of leader and moderator. These sessions encompassed an educational element, featuring theoretical and practical presentations related to breastfeeding and aligned with the recommendations of the Baby-Friendly Hospital Initiative [39]. They also included motivational and social or peer support components established within the group. Consequently, on a monthly basis, women received support from an organised and proactive professional. In addition to these monthly gatherings, participants had the opportunity to interact with each other, connect with other breastfeeding women and communicate with the designated midwife through a Facebook™ and/or WhatsApp™ group specifically created for this purpose. This strengthened peer support, and queries regarding the topic were addressed using information and communication technologies [40]. Similarly, participating women retained the option to request individual consultations with the designated midwife on demand, similar to those receiving standard care.

### 2.7. Assessment

Sociodemographic and obstetric clinical data were collected by a questionnaire designed for this purpose via a web application. Incorrect or incomplete data were corrected via direct consultation with participants or were collected from their medical records with their consent. The data collected included the following:Sociodemographic variables: maternal age, country of origin, civil status (single, married, separated, widow), educational level (none, primary school, secondary school, university), employment status (self-employed, employed, unemployed);Obstetric variables: parity (primiparous, multiparous), gestational age, labour onset (induction, spontaneous), type of birth (eutocic, instrumental, elective caesarean section, emergent caesarean section), newborn sex, birth weight.

The type of breastfeeding was recorded at hospital discharge, as well as at three established follow-up time points: 10 days postpartum (T1), 2 months postpartum (T2) and 4 months postpartum (T3). Distinctions were made between EBF, breastfeeding with occasional supplementation of formula, mixed feeding and formula feeding.

PPD was measured using the Edinburgh Postnatal Depression Scale (EPDS) designed by Cox et al. [41] in 1987 and validated for the Spanish population by García-Esteve et al. [42] in 2003. This is a 10-item self-reported scale in which women indicate how they felt in the last 7 days. The scale is structured into three factors: anhedonia (items 1, 2 and 10), anxiety (items 3–6) and depressive symptomatology (items 7–9) [43]. The minimum possible score is 0, and the maximum is 30. The best cut-off of the Spanish validation of the EPDS was 10/11 for combined major and minor depression, the sensitivity was 79% and the specificity was 95.5%, with a positive predictive value of 63.2% and a negative predictive value of 97.7%. At this cut-off, all cases of major depression were detected. The area under the receiver operating characteristic curve was 0.976 (*p* = 0.001) with an asymptotic 95% confidence interval between 0.968 and 0.984 [42].

General self-efficacy was measured using the General Self-efficacy Scale (GSE) designed by Baessler and Schwarcer [44] in 1996. It was validated for the Spanish population by Sanjuán et al. [45]. This scale assesses the enduring sense of personal competence to effectively handle a wide variety of stressful situations. It is a unidimensional scale with 10 Likert-type questions [44]. A change in the original response form (10-point Likert-type scale instead of a 4-point scale) was introduced in order to adapt the scale to other research instruments. The reliability of the Spanish version of the GSE, as measured by the Cronbach alpha coefficient, was 0.87 [45].

The main control and outcome variables were measured before the start of the intervention (baseline) and at 2- and 4-month follow-ups.

### 2.8. Data Collection

The enrolment of participants commenced in October 2021 and concluded in May 2023. This process was performed by the midwives overseeing each health centre. These midwives underwent prior training for the project and received guidance from a research technician midwife associated with the project but not directly involved in the intervention. The designated midwife at the health centre, during consultations with eligible women, provided information about the study’s nature and objectives, as well as details regarding the follow-up procedures. Once participants provided information via the project’s web application, they agreed to participate and signed the informed consent form in duplicate. The web application automatically sent them reminder messages and emails at the three evaluation time points established in the study.

The data relating to electronic follow-up were coded and safeguarded by the research team. All data were stored in an electronic database accessible only to members of the research team.

### 2.9. Data Analysis

Descriptive data analyses were conducted to characterise the variables. Baseline characteristics were compared between the group experiencing potential losses during follow-up and the group completing follow-up using cross-tabulation analysis. Means were compared using Fisher’s exact or *t*-tests, as appropriate. Associations between baseline and childbirth variables and EBF maintenance at 10 days, 2 months and 4 months postpartum were examined using cross-tabulation analysis.

A per-protocol analysis was performed. Chi-square or Fisher’s exact tests and ANOVA or *t*-tests, as appropriate, were employed for mean comparisons. To assess the effect of the intervention on EBF maintenance at various postpartum time points, cross-tabulation analysis and chi-square tests were utilised. Additionally, a multivariate logistic model was employed to calculate adjusted odds ratios and their 95% confidence intervals for each time point.

The assumption that variables were normally distributed was checked using the Kolmogorov–Smirnov test. Group homogeneity analyses based on baseline and childbirth variables were conducted using cross-tabulation analysis, utilising chi-square or Fisher’s exact tests as needed. ANOVA and *t*-tests were employed for mean comparisons.

Data analysis was conducted using SPSS v. 28.1 for Windows (IBM Corp. 2018, Armonk, NY, USA) and R (R Project 2019, version 4.0.2). The threshold for statistical significance was set at *p* < 0.05.

### 2.10. Ethical Considerations

Before beginning the study, it was approved by the Research Ethics Committees of the Virgen Macarena and Virgen del Rocío hospitals (Seville, Spain) on 13 March 2021 (Code 2722-N-20).

Participation in the project was voluntary, as was the participation request. Verbal and written informed consent information was provided to every participant in the study. The study was designed according to Spanish Law No. 14/2007 of 3 July regarding biomedical research and complied with the study suitability requirements and with the procedure regarding the study objectives. The data were anonymously handled according to the Spanish Organic Law on Protection of Personal Data and Guarantee of Digital Rights (Spanish Organic Law 3/2018).

## 3. Results

### 3.1. Characteristics of the Sample

A total of 512 participants were initially selected, with 130 (25.4%) excluded from randomisation for the following reasons: 73 (56.2%) were not breastfeeding their newborns and 57 (43.8%) declined follow-up in the first 10 days postpartum.

The analysis focused on a total sample of 382 mother–child dyads, randomly distributed, with 151 (39.5%) in the CG and 231 (60.5%) in the IG. There were 51 (13.35%) dropouts between T1 and T2 (*n* = 331), 27 (7.06%) of them due to discontinuation of breastfeeding. In addition, 28 participants (7.32%) dropped out between T2 and T3 (*n* = 303), motivated by discontinuation of breastfeeding, resulting in a total of 79 participants who did not continue responding to surveys (Figure 1).

We compared baseline characteristics between the dropout group (*n* = 79 [CG: 29; IG: 50]) and the final analysed group (*n* = 303). Fisher’s exact and *t*-tests were used, as appropriate for variable types, to compare the groups. We observed that only those women in the IG dropout group had a lower rate of university education compared to the follow-up group (52.0% vs. 66.9%); this difference was statistically significant (*p* = 0.038). Thus, despite these losses, group homogeneity was maintained, indicating their random origin.

### 3.2. Sociodemographic and Obstetric–Neonatal Variables

The participants had a mean age of 33.4 ± 4.7 years, with 93.5% (357/382) born in Spain. The majority had a university education (64.4%), were married (55.0%) and had gainful employment (61.5%). The mean gestational age at birth was 39.5 ± 1.2 weeks, and 53.9% (206/382) of participants were primiparous, with 60.7% (232/382) experiencing a spontaneous onset of labour culminating in vaginal delivery (61.8%). The average birth weight was 3271 ± 434.3 g.

The relative rate of breastfeeding experience was 38.4% (58/151) in the CG and 44.6% (103/231) in the IG. We did not find statistically significant differences between the sociodemographic or obstetric–neonatal characteristics of the two groups, except for early skin-to-skin contact (*p* = 0.028) and feeding type at 4 months (*p* < 0.001; Table 1).

During the follow-up period, we observed a gradual reduction in the breastfeeding rate from 78.0% (298/382) at 10 days to 69.5% (230/331) at 2 months and 62.4% (189/303) at 4 months postpartum. Statistically significant differences were found between the rates of breastfeeding in the CG (50.0%) and the IG (70.7%) at 4 months postpartum (*p* < 0.001; Table 2).

Statistically significant differences between the groups were observed in PPD at 4 months postpartum, with a lower mean score on the EPDS in the IG than the CG (12.49 ± 3.6 vs. 13.39 ± 4.0; *p* = 0.044; Table 3).

We examined the relationships between the maintenance of EBF and both EPDS and GSE scores during the study period. We observed statistically significant differences in the GSE scores of women who did and did not perform EBF only at T2 and T3, with women performing EBF obtaining higher scores (78.1 ± 14.3 vs. 74.3 ± 15.2 at T2 [*p* = 0.014]; 78.3 ± 14.4 vs. 72.4 ± 15.9 at T3 [*p* < 0.001]; Table 4). Statistically significant differences were observed in the EPDS scores of women who did and did not perform EBF only at T2 and T3, with lower mean scores in women performing EBF (12.2 ± 3.5 vs. 13.5 ± 3.9 at T2 [*p* = 0.002]; 12.1 ± 3.6 vs. 14.1 ± 3.8 at T3 [*p* < 0.001]; Table 4).

Table 5 presents the factors associated with the maintenance of EBF across the three distinct postpartum periods. Logistic regression analysis results revealed significant associations between various variables and the likelihood of sustaining EBF during each period. At T1, the absence of early skin-to-skin contact was significantly associated with a decrease in the likelihood of maintaining EBF (OR = 0.432, *p* = 0.014). At T2, EPDS scores were significantly associated with the likelihood of maintaining EBF. Specifically, an increase in EPDS T2 scores was linked to a significant decrease in the likelihood of sustaining EBF (OR = 0.915, *p* = 0.012). This finding suggests that higher levels of depressive symptoms during the second postpartum period were associated with a reduction in the likelihood of maintaining EBF. Similarly, at T3, EPDS scores were significantly associated with the probability of maintaining EBF, with an increase in score linked to a significant decrease in the likelihood of sustaining EBF (OR = 0.887, *p* = 0.002). Finally, the absence of intervention was related to a significant decrease in the probability of maintaining EBF at T3 (OR = 0.474, *p* = 0.003).

## 4. Discussion

The purpose of this study was to assess the effect of midwife-led breastfeeding support groups on the maintenance of breastfeeding, the prevalence of PPD and the perceived general self-efficacy of the participants.

In our study, one of the most important factors related to the initiation of breastfeeding was early skin-to-skin contact after delivery. This result aligns with findings from a Cochrane review indicating that this intimate contact between the newborn and the mother provides a unique environment that meets basic biological needs, according to mammalian neuroscience, and programs future behaviours that aid in the maintenance of EBF [46]. Breastfeeding is considered a protective factor against PPD because it causes the release of oxytocin, which contributes to the well-being of the woman [47].

Another factor in our study related to the initiation of breastfeeding was greater gestational age at birth, indicating that these newborns had greater biological maturity that allowed for a more satisfactory initiation of breastfeeding and better adaptation to extrauterine life. Conversely, early term infants (born between weeks 37 + 0 and 38 + 6) are more likely to experience adverse neonatal outcomes that necessitate medical interventions, thereby complicating the initiation of breastfeeding [48,49]. This could not be analysed in our study, as participants with preterm pregnancies were not included.

The maintenance of breastfeeding during the first 6 months plays a crucial role in the health and well-being of the mother–infant dyad. According to our data on postpartum depressive symptoms, as measured by the EPDS, higher levels of depressive symptoms were associated with a reduction in the maintenance of EBF at 2 months (T2) and 4 months (T3). However, we must clarify that the direction of the association is unknown, as we do not know whether women who report fewer signs and symptoms of PPD have better breastfeeding experiences or whether those who continue breastfeeding for a longer period adapt more effectively postpartum and therefore have lower PPD scores. This challenge has already been identified by other authors who reported that women who breastfed for a longer duration had a statistically significantly lower EPDS risk score for PPD [50]. A study by Bascom et al. [51] suggested that, when depressive symptoms appear in postpartum women prematurely, difficulties with breastfeeding often lead to its early cessation.

Another key factor for the maintenance of breastfeeding is education and support through breastfeeding support groups. Our findings align with those of other studies, which have shown that interventions for promoting breastfeeding based on a combination of social support from peers and leadership by IBCLCs yield better results in maintaining breastfeeding during the first 6 months postpartum [29]. In our study, we did not observe differences at 2 months postpartum, when the first breastfeeding challenge occurs [52], but we did observe differences at 4 months postpartum, when women return to work, as indicated by other studies in which interventions were effective at 4 and 6 months postpartum [53]. The national regulation for maternity leave in Spain in relation to the workers’ statute and public employees is 16 weeks with the following distribution: 6 mandatory interrupted weeks that must be enjoyed full-time immediately after giving birth and 10 more weeks that can be enjoyed on a full- or part-time basis [54]. This period of time aligns with the return to work.

General self-efficacy in breastfeeding, which is based on confidence, helps to improve breastfeeding rates [55]. In our study, it was linked to breastfeeding and PPD. Women who demonstrated higher general self-efficacy showed higher levels of breastfeeding [56] and lower levels of PPD. Additionally, those who participated in breastfeeding support groups had better outcomes for the aforementioned parameters. This aligns with the findings of Tsen et al. [57] in their randomised controlled trial, which indicate that previous breastfeeding experiences (performance accomplishments), along with observing successful breastfeeding in peers (vicarious experience) and verbal encouragement from a leader promoting breastfeeding (verbal persuasion), lead to breastfeeding success. The stress and anxiety reduction provided by these support groups increases self-efficacy and, consequently, breastfeeding [57]. Additionally, our study showed that women who were part of midwife-led breastfeeding support groups maintained breastfeeding for a longer duration and experienced less PPD. Hence, multiple findings suggest that support groups have numerous benefits as a health promotion strategy and coping mechanism for illnesses through informative support, shared experiences and opportunities to learn from others [58,59].

We must acknowledge some limitations of our study. Firstly, in the control group, no intervention unrelated to breastfeeding was carried out, but it did include psychosocial care. The reason was that in this study, we compared the standard care that women receive during this period [38] with an additional intervention designed primarily to improve breastfeeding rates. However, we also presumed that it could potentially enhance maternal mental health. Furthermore, since the direction of the association between breastfeeding and postpartum depression is uncertain, implementing an emotional support intervention in the control group could indirectly improve breastfeeding rates and confound the results. Nevertheless, women in the control group had access to on-demand consultations with the midwife for assistance or advice if needed. Secondly, only healthy women and newborns were included, thereby limiting the variability of observed physiological parameters and potentially simplifying their interpretation. Data related to breastfeeding type were self-reported, which could introduce a memory bias, even though data collection was conducted chronologically over time. Additionally, pregnant women might have misclassified types of breastfeeding (EBF, breastfeeding with other foods, etc.), which could also be related to memory bias. However, maternal recall for reporting these data has been shown to be a valid and reliable estimate of breastfeeding [60]. Furthermore, no data were collected on informal support or on the partners of the participants (age, gender, education, occupation…). Thirdly, we lacked a baseline assessment prior to pregnancy of women’s rates of depression. We cannot determine if women who experienced PPD had previously suffered from depression or showed signs of being at risk. Another limitation was the withdrawal of patients from both the CG and the IG throughout the period of data collection. Finally, follow-up was conducted only until 4 months postpartum, as it is reported to be a significant period for early discontinuation of breastfeeding, particularly due to work-related reasons. In subsequent studies, it would be relevant to address up to 6 months postpartum, as per the World Health Organization’s recommendation for breastfeeding.

We would like to highlight several strengths of our study, such as the prospective, consecutive and randomised inclusion of patients in five provinces of Andalusia, a southern region of Spain, allowing the findings to be applicable in routine clinical practice. The training provided by the research team to the midwives recruiting pregnant women ensured that the CG and IG samples were as homogeneous as possible. Another significant strength of this study was the use of multivariate logistic regression to determine the factors favouring breastfeeding at different data collection time points (from T1 to T3), a statistical approach not implemented in previous studies.

## 5. Conclusions

Women participating in midwife-led breastfeeding support groups exhibited higher levels of general self-efficacy, maintained breastfeeding for a longer duration and showed less PPD at 4 months after childbirth compared to women in the CG. These findings suggest the need for healthcare providers (midwives) to develop intervention strategies that address factors supporting the initiation and maintenance of breastfeeding by enhancing self-efficacy to reduce the occurrence of PPD, as these have been identified as promising interventions, although further research is needed.

## Figures and Tables

**Figure 1 nutrients-16-00227-f001:**
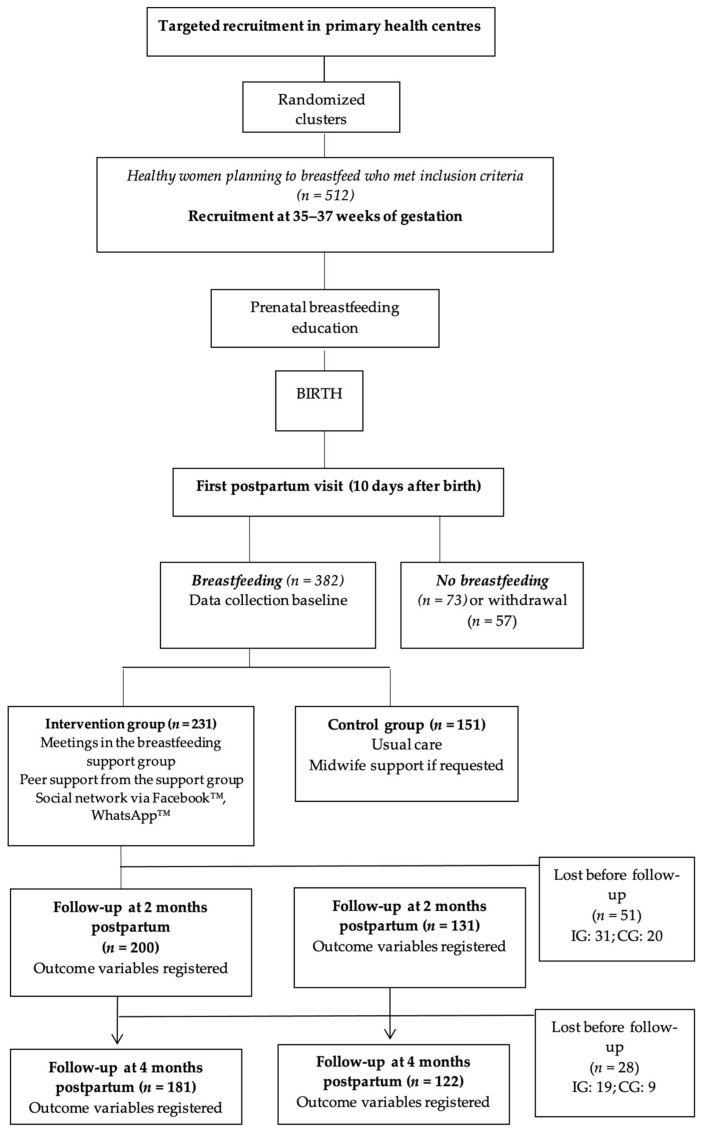
Participant selection flowchart.

**Table 1 nutrients-16-00227-t001:** Distributions of baseline variables in control and intervention groups (*n* = 382).

		Control Group *n* = 151 (39.5%)		Intervention Group *n* = 231 (60.5%)		*p*-Value *
		*n*	%	*n*	%	
Country of origin	Spain	142	94	215	93.1	0.709
	Foreign	9	6	16	6.9	
Civil status	Single	73	48.3	95	41.1	0.326
	Married	76	50.3	134	58	
	Separate	2	1.3	2	0.9	
	Widow	0	0	0	0	
Educational level	None	0	0	1	0.4	0.846
	Primary school	5	3.3	9	3.9	
	Secondary school	47	31.1	74	32	
	University	99	65.6	147	63.6	
Employment status	Self-employed	11	7.3	27	11.7	0.353
	Employed	97	64.2	138	59.7	
	Unemployed	43	28.5	66	28.6	
Parity	Primiparous	87	57.6	119	51.5	0.242
	Multiparous	64	42.4	112	48.5	
Previous BF experience	No	93	61.6	128	55.4	0.232
	Yes	58	38.4	103	44.6	
Labour onset	Induction	61	40.4	89	38.5	0.715
	Spontaneous	90	59.6	142	61.5	
Type of birth	Eutocic	96	63.6	140	60.6	0.411
	Instrumental	26	17.2	51	22.1	
	Elective CS	5	3.3	12	5.2	
	Emergent CS	24	15.9	28	12.1	
E-SSC	No	25	16.6	21	9.1	0.028
	Yes	126	83.4	210	90.9	
Newborn sex	Male	79	52.3	116	50.2	0.688
	Female	72	47.7	115	49.8	
Type of feeding at discharge (*n* = 382)	EBF	121	80.1	178	77.1	0.841
	BF with OH	17	11.3	32	13.8	
	Mixed	13	8.6	21	9.1	
	Formula	-	-	-	-	
Type of feeding T1 (*n* = 382)	EBF	118	78.1	180	77.9	0.960
	BF with OH	20	13.3	31	13.4	
	Mixed	13	8.6	20	8.7	
	Formula	-	-	-	-	
Type of feeding T2 (*n* = 331)	EBF	84	64.1	146	73	0.335
	BF with OH	14	10.7	18	9	
	Mixed	19	14.5	23	11.5	
	Formula	14	10.7	13	6.5	
Type of feeding T3 (*n* = 303)	EBF	61	50	128	69.9	<0.001
	BF with OH	13	10.7	21	11.60	
	Mixed	22	18	12	6.62	
	Formula	26	21.3	20	10.9	
Quantitative Variables	Group	*n*	Mean		SD	*p*-value **
Maternal age (year)	CG	151	33.28		5.03	0.063
	IG	231	33.50		4.41	
Gestational age (week)	CG	151	39.46		1.38	0.820
	IG	231	39.45		1.14	
Birth weight (gram)	CG	151	3299		430	0.819
	IG	230	3253		437	
EPDS T1 (*n* = 382)	CG	151	12.65		3.68	0.090
	IG	231	12.11		3.26	
EPDS T2 (*n* = 331)	CG	131	12.50		3.66	0.487
	IG	200	12.62		3.70	
EPDS T3 (*n* = 303)	CG	122	13.39		4.00	0.116
	IG	181	12.49		3.63	
GSE T1 (*n* = 382)	CG	151	78.59		14.36	0.699
	IG	231	79.58		13.87	
GSE T2 (*n* = 331)	CG	131	75.65		14.39	0.607
	IG	200	77.73		14.81	
GSE T3 (*n* = 303)	CG	122	75.36		15.17	0.881
	IG	181	76.46		15.26	

* Chi-squared test; ** ANOVA; BF: breastfeeding; CS: caesarean section; E-SSC: early skin-to-skin contact; EBF: exclusive breastfeeding; BF with OH: breastfeeding with occasional help; T1: 10 days postpartum; T2: 2 months postpartum; T3: 4 months postpartum; SD: standard deviation; EPDS: Edinburg Postnatal Depression Scale; GSE: General Self-efficacy Scale; CG: control group; IG: intervention group.

**Table 2 nutrients-16-00227-t002:** Analysis of the between-group differences in the maintenance of exclusive breastfeeding.

			Group		Total	*p*-Value *
			CG	IG		
EBF T1 (*n* = 382)	No	*n*	33	51	84	0.959
		%	21.90	22.10	22.00	
	Yes	*n*	118	180	298	
		%	78.10	77.90	78.00	
EBF T2 (*n* = 331)	No	*n*	47	54	101	0.086
		%	35.90	27.00	30.50	
	Yes	*n*	84	146	230	
		%	64.10	73.00	69.50	
EBF T3 (*n* = 303)	No	*n*	61	53	114	<0.001
		%	50.00	29.28	37.62	
	Yes	*n*	61	128	189	
		%	50.00	70.72	62.38	

* Chi-square test; CG: control group; IG: intervention group; EBF: exclusive breastfeeding; T1: 10 days postpartum; T2: 2 months postpartum; T3: 4 months postpartum.

**Table 3 nutrients-16-00227-t003:** Effectiveness of the intervention at reducing postpartum depression, as evidenced by between-group differences.

		*n*	Mean	SD	95% CI		Minimum	Maximum	*F*	*p*-Value *
					Upper Limit	Lower Limit				
EPDS T1	CG	151	12.65	3.686	12.06	13.24	6	23	2.258	0.134
	IG	231	12.11	3.268	11.68	12.53	6	23		
	Total	382	12.32	3.445	11.98	12.67	6	23		
EPDS T2	CG	131	12.50	3.666	11.87	13.14	6	24	0.072	0.789
	IG	200	12.62	3.702	12.10	13.13	6	22		
	Total	331	12.57	3.683	12.17	12.97	6	24		
EPDS T3	CG	122	13.39	4.001	12.67	14.10	6	23	4.077	0.044
	IG	181	12.49	3.636	11.96	13.02	6	24		
	Total	303	12.85	3.805	12.42	13.28	6	24		

* ANOVA; SD: standard deviation; CI: confidence interval; EPDS: Edinburg Postnatal Depression Scale; T1: 10 days postpartum; T2: 2 months postpartum; T3: 4 months postpartum; CG: control group; IG: intervention group.

**Table 4 nutrients-16-00227-t004:** Relationships between the maintenance of EBF and both EPDS and GSE scores during the study period.

							Levene’s Test for Equality of Variances		*t*-Test for Equality of Means							
							*F*	*p*-Value	*t*	*df*	Significance		Mean Difference	Standard Error of Difference	95% CI	
			*n*	Mean	SD	Standard Error of the Mean					One-Way *p*-Value	Two-Way *p*-Value			Lower	Upper
EBF T1	EPDS T1	No	84	12.4	3.1	0.3	1.006	0.317	0.285	380	0.388	0.776	0.121	0.426	−0.716	0.959
		Yes	298	12.3	3.5	0.2										
	GSE T1	No	84	80.2	12.1	1.3	1.799	0.181	0.744	380	0.229	0.457	1.293	1.738	−2.123	4.709
		Yes	298	78.9	14.6	0.8										
EBF T2	EPDS T2	No	101	13.5	3.9	0.4	2.180	0.141	3.165	329	0.001	0.002	1.373	0.434	0.519	2.226
		Yes	230	12.2	3.5	0.2										
	GSE T2	No	101	74.3	15.2	1.5	0.281	0.596	−2.202	329	0.014	0.028	−3.831	1.740	−7.253	−0.408
		Yes	230	78.1	14.3	0.9										
EBF T3	EPDS T3	No	116	14.1	3.7	0.3	0.023	0.880	4.656	303	<0.001	<0.001	2.015	0.433	1.163	2.867
		Yes	187	12.1	3.6	0.3										
	GSE T3	No	116	72.4	15.9	1.5	1.121	0.291	−3.318	303	<0.001	<0.001	−5.587	1.809	−9.331	−2.384
		Yes	187	78.3	14.4	1.0										

EBF: Exclusive breastfeeding; T1: 10 days postpartum; T2: 2 months postpartum; T3: 4 months postpartum; EPDS: Edinburg Postnatal Depression Scale; GSE: General Self-efficacy Scale.

**Table 5 nutrients-16-00227-t005:** Multivariate logistic regression models of factors favouring exclusive breastfeeding during the study period.

		β	Standard Error	Wald	*df*	*p*-Value	Exp (β)	95% CI	
								Lower	Upper
T1	E-SSC (No)	−0.839	0.341	6.047	1.000	0.014	0.432	0.221	0.843
	Gestational age	0.164	0.097	2.866	1.000	0.090	1.178	0.975	1.424
	EPDS T1	−0.011	0.036	0.085	1.000	0.770	0.989	0.921	1.063
	GSE T1	−0.007	0.009	0.591	1.000	0.442	0.993	0.975	1.011
	Intervention (No)	0.083	0.259	0.102	1.000	0.749	1.086	0.653	1.807
	Constant	−4.392	3.924	1.252	1.000	0.263	0.012		
T2	E-SSC (No)	−0.333	0.376	0.781	1.000	0.377	0.717	0.343	1.499
	Gestational age	0.043	0.101	0.177	1.000	0.674	1.044	0.856	1.273
	EPDS T2	−0.089	0.035	6.315	1.000	0.012	0.915	0.853	0.981
	GSE T2	0.009	0.009	1.028	1.000	0.311	1.009	0.992	1.026
	Intervention (No)	−0.403	0.249	2.612	1.000	0.106	0.668	0.410	1.090
	Constant	−0.187	4.044	0.002	1.000	0.963	0.829		
T3	E-SSC (No)	−0.737	0.397	3.438	1.000	0.064	0.479	0.220	1.043
	Gestational age	0.069	0.105	0.430	1.000	0.512	1.072	0.872	1.317
	EPDS T3	−0.120	0.039	9.406	1.000	0.002	0.887	0.822	0.958
	GSE T3	0.011	0.010	1.189	1.000	0.275	1.011	0.992	1.030
	Intervention (No)	−0.746	0.255	8.566	1.000	0.003	0.474	0.288	0.782
	Constant	−1.076	4.292	0.063	1.000	0.802	0.341		

Adjusted *R*^2^ for T1: 0.219; adjusted *R*^2^ for T2: 0.472; adjusted *R*^2^ for T3: 0.37; Exp (β): odds ratio; T1: 10 days postpartum; T2: 2 months postpartum; T3: 4 months postpartum; E-SSC: early skin-to-skin contact; EPDS: Edinburg Postnatal Depression Scale; GSE: General Self-efficacy Scale.

## Data Availability

The data presented in this study are available on request from the corresponding author. The data are not publicly available due to confidentiality issues.

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
