# Peer review of "Evaluation of the Impact of a Midwife-Led Breastfeeding Group Intervention on Prevention of Postpartum Depression: A Multicentre Randomised Clinical Trial"

_nutrients, 2024, doi:10.3390/nu16020227_

Round 1
Reviewer 1 Report
Comments and Suggestions for Authors
Evaluation of the impact of a midwife-led breastfeeding group intervention on prevention of postpartum depression: a multicentre randomized clinical trial
This study conducted in Andalusia, Spain, from October 2021 to May 2023 aimed to assess the effectiveness of a midwife-led breastfeeding support group intervention on breastfeeding rates, postpartum depression, and general self-efficacy. The multicentric cluster randomized controlled trial involved 382 women and revealed several significant findings: Exclusive Breastfeeding Rates: There was a significant difference in exclusive breastfeeding rates at 4 months postpartum between the intervention and control groups, with the intervention group showing higher rates (p < 0.001). Postpartum Depression: The intervention group exhibited a lower mean score on the Edinburgh Postnatal Depression Scale compared to the control group (12.49 ± 3.6 vs. 13.39 ± 4.0; p = 0.044). This suggests that the midwife-led breastfeeding support group intervention had a positive impact on reducing postpartum depression. General Self-Efficacy: Breastfeeding women in the intervention group demonstrated higher scores of general self-efficacy at 2 and 4 months postpartum compared to the control group (p = 0.002 and p < 0.001, respectively). This indicates that the intervention contributed to an increased sense of self-efficacy among participants. In conclusion, the midwife-led breastfeeding support groups proved to be effective in enhancing self-efficacy, prolonging breastfeeding, and reducing postpartum depression four months after giving birth. These findings highlight the potential benefits of group support interventions in addressing postpartum mental health issues and promoting positive breastfeeding outcomes.
Title: The title is well chosen, reflecting the study being reported.
Overall: The aim is well emphasized and explained. The paper is well written and was very pleasant to read.
Abstract: The abstract is written correctly. However, please provide specific values in the abstract and not just significance levels.
Introduction :
The introduction section is attractive to read, emphasizing the reason for conducting this study. The background of mental symptoms is explained. However, I propose to deepen the information on the background and mechanisms underlying postpartum depression. The role of unstable progesterone levels in maintaining normal mood in other physiological and pathological states is also worth to outline. You can deepen the description based on the work of 37111278.
It would also be appropriate in the introduction to briefly describe what is the effect of breastfeeding on hormone secretion and its direct impact on mood disorders.
Material and Methods
This section is well written, explaining the method the study was conducting. Despite, there are several important and crucial points needing explanation and/or clarification:
1.”Participants and Study Area” - Instead of giving the exact number of inhabitants (to the nearest 1 person - the number is certainly variable and approximate, and adds nothing to the work), it would be useful to give what is the fertility rate of women in the population and the average age of birth of the first child.
2. “Exclusion criteria included the following”: - the dot is unnecessary
Results
This section is well written, explaining the outcomes of the study.
Discussion
This section is well written and very pleasant to read. But:
Additional limitations of the study would have to be described. For example, why the control group was not subjected to a similar interaction with the instructor that did not involve breastfeeding, but provided similar social-psychological interference: contact with the midwife, messenger groups, Facebook groups.
If the authors link the issue of breastfeeding to the return to work, it would have been necessary to show what local regulations and practices on this issue are.
Similarly, no data was included (not collected), any data on the partners of patients: age, education, salary, etc. We know from our own data that such variables can affect breastfeeding in a significant way. The discussion section should be adapted thereafter.
Author Response
Dear Editors and Reviewers,
Thank you for reviewing the manuscript and for your comments. Please find below in the text the response to each comment. All the changes in the manuscript have been modified in red.
REVIEWER 1:
Title: The title is well chosen, reflecting the study being reported.
Overall: The aim is well emphasized and explained. The paper is well written and was very pleasant to read.
Abstract: The abstract is written correctly. However, please provide specific values in the abstract and not just significance levels.
Response: Thank you for reviewing the manuscript and for your feedback to improve it. New values have been included in the summary marked in red.Please, see lines 20-21, 28, 31-32.
Introduction :
The introduction section is attractive to read, emphasizing the reason for conducting this study. The background of mental symptoms is explained. However, I propose to deepen the information on the background and mechanisms underlying postpartum depression. The role of unstable progesterone levels in maintaining normal mood in other physiological and pathological states is also worth to outline. You can deepen the description based on the work of 37111278.
It would also be appropriate in the introduction to briefly describe what is the effect of breastfeeding on hormone secretion and its direct impact on mood disorders.
Response: Thank you for your feedback. A brief explanation is added based on the suggested article. These clarifications are added in red. Please, see lines 58-69.
Material and Methods
This section is well written, explaining the method the study was conducting. Despite, there are several important and crucial points needing explanation and/or clarification:
1.”Participants and Study Area” - Instead of giving the exact number of inhabitants (to the nearest 1 person - the number is certainly variable and approximate, and adds nothing to the work), it would be useful to give what is the fertility rate of women in the population and the average age of birth of the first child.
Response: The data regarding the total population have been replaced by the rate of women of reproductive age and average age of birth of the first child. Changes are marked in red. Please, see lines 116-118.
- “Exclusion criteria included the following”: - the dot is unnecessary
Response: Has been modified. Please, see line 126.
Results
This section is well written, explaining the outcomes of the study.
Discussion
This section is well written and very pleasant to read. But:
Additional limitations of the study would have to be described. For example, why the control group was not subjected to a similar interaction with the instructor that did not involve breastfeeding, but provided similar social-psychological interference: contact with the midwife, messenger groups, Facebook groups.
Response: The main objective of the study was to track breastfeeding, not to prevent postpartum depression. We simply wanted to see the possible relationship between participation in breastfeeding support groups, the maintenance of breastfeeding, and postpartum depression. However, some text was included. Please, see lines 429-438.
If the authors link the issue of breastfeeding to the return to work, it would have been necessary to show what local regulations and practices on this issue are.
Response: Thank you for your comment. A brief explanation of the duration of the maternity leave in Spain is added in red. Please, see lines 408-412.
Similarly, no data was included (not collected), any data on the partners of patients: age, education, salary, etc. We know from our own data that such variables can affect breastfeeding in a significant way. The discussion section should be adapted thereafter.
Response: Thank you for reviewing this important section of the manuscript and for your feedback to improve it. New clarifications have been added in red. Please, see lines 445-447.
Reviewer 2 Report
Comments and Suggestions for Authors
The manuscript by Gallego et al. assessed the effectiveness of a midwife-led breastfeeding support group intervention on breastfeeding rates, postpartum depression and general self-efficacy. Given the high prevalence rate and serious morbidity of PPD, this study was of high significance. However, to enhance the manuscript, the following modifications are suggested:
1. The child-birth history, previously diagnosed PPD, the stress level from work and life should be evaluated.
2. Are there any associations between the incidence of PPD and the family history of the disease? Are there any possible gene mutations that could account for the incidence of PPD?
3. Why was 4 month selected instead of 6 month?
Author Response
Dear Editors and Reviewers,
Thank you for reviewing the manuscript and for your comments. Please find below in the text the response to each comment. All the changes in the manuscript have been modified in red.
REVIEWER 2:
The manuscript by Gallego et al. assessed the effectiveness of a midwife-led breastfeeding support group intervention on breastfeeding rates, postpartum depression and general self-efficacy. Given the high prevalence rate and serious morbidity of PPD, this study was of high significance. However, to enhance the manuscript, the following modifications are suggested:
- The child-birth history, previously diagnosed PPD, the stress level from work and life should be evaluated.
Response: Thank you for this observation, however, it was not the objective of the study to delve into postpartum depression in such detail. It is very interesting to consider these values for future research.
- Are there any associations between the incidence of PPD and the family history of the disease? Are there any possible gene mutations that could account for the incidence of PPD?
Response: Thank you for your reflection on the topic. Some studies associate family history with the risk of developing postpartum depression, such as the systematic review and meta-analysis of Zacher Kjeldsen et al (2022). Also, there is some evidence regarding certain genetic mutations that are associated with the development of this pathology in the postpartum period (Khabour et al, 2013). However, it was not the objective of the present study, so it was not considered as a factor in the presentation of results. The sample was randomized to avoid this bias.
Zacher Kjeldsen MM, Bricca A, Liu X, Frokjaer VG, Madsen KB, Munk-Olsen T. Family History of Psychiatric Disorders as a Risk Factor for Maternal Postpartum Depression: A Systematic Review and Meta-analysis. JAMA Psychiatry. 2022 Oct 1;79(10):1004-1013. doi: 10.1001/jamapsychiatry.2022.2400.
Khabour O, Amarneh B, Bani Hani E, Lataifeh I. Associations Between Variations in TPH1 , TPH2 and SLC6A4 Genes and Postpartum Depression: A Study in the Jordanian Population. Balkan J Med Genet. 2013 Jun;16(1):41-8. doi: 10.2478/bjmg-2013-0016.
- Why was 4 month selected instead of 6 month?
Response: This initial study has a time cut-off at 4 months, as it aligns with the return to work period following paid leave in Spain, which is reported as one of the main reasons for early discontinuation of breastfeeding. In future research, it would be advisable to extend the time frame to 6 months. This clarification has been made in the discussion, highlighted in red. Please, see lines 451-455.